# Benchmarking Refined Quantum Linear Systems Algorithms

Adrian Harkness[1][0009−0001−5518−6442], Kate Saltovets[1], Mohammadhossein Mohammadisiahroudi[1][0000−0002−4046−0672], and Tamás Terlaky[1][0000−0003−1953−1971]

Industrial and Systems Engineering, Lehigh University, Bethlehem, PA 18015, USA

**Abstract.** Systems of linear equations are ubiquitous across science, engineering, machine learning, and even finance. While classical methods can be prohibitively slow for large-scale problems, quantum linear systems algorithms offer the potential for exponential speedup in certain parameter regimes. However, a significant gap persists between this theoretical promise and practical implementation, as the advantages are often obscured by the substantial quantum resources and high sensitivity to noise inherent in current quantum hardware. One way to bridge this gap is through the use of Iterative Refinement, a classical post-processing scheme that can exponentially improve the accuracy to which a linear system of equations can be solved using low-precision arithmetic. In the context of quantum linear systems algorithms, such as the HHL algorithm proposed by Harrow, Hassidim, and Loyd, Iterative Refinement can greatly reduce the quantum resources required to calculate an accurate solution in terms of tomography cost, circuit volume, and fault-tolerant overhead. Here, we compute and benchmark highly precise solutions to linear systems of equations of up to eight variables by running HHL with Iterative Refinement on NISQ quantum computers. We also present our open-source implementation, emphasizing that our circuit is not tailored to specific problem instances, as most available implementations are.

**Keywords:** Quantum Linear System Algorithm · HHL Algorithm · Iterative Refinement · Benchmarking.

## 1 Introduction

It is widely believed that quantum computers can efficiently solve some problems that do not admit efficient classical algorithms. Recently, a considerable amount of attention has been devoted to quantum linear algebra. A common class of methods use Hamiltonian simulation to prepare a quantum state $|x\rangle$ that is proportional to the solution of the linear system

$$Ax = b, \tag{1}$$

for a given $s$-row sparse Hermitian matrix $A \in \mathbb{R}^{n \times n}$ (i.e., $A = A^\dagger$ has at most $s$ nonzero entries per row) and a vector $b \in \mathbb{R}^n$. In the *quantum linear systems*

*problem* (QLSP), one has input oracle access to the entries of $A$ and the ability to prepare a quantum state $|b\rangle$ that is proportional to the right hand side vector $b$. It is also standard to assume that $\|A\| \leq 1$, and that its non-zero eigenvalues lie in $[-1, -1/\kappa_A] \cup [1/\kappa_A, 1]$, where $\kappa_A$ the (known) condition number of $A$.

Research into this subfield began with the work of Harrow, Hassidim, and Lloyd [4], who proposed what has come to be known as the HHL algorithm for solving the QLSP. In this seminal work, it was shown that a quantum computer could be used to solve a QLSP with a worst-case complexity of

$$\mathcal{O}\left(\frac{s^2 \kappa_M^2}{\epsilon} \cdot \mathrm{polylog}(d)\right),$$

where $\epsilon > 0$ is the accuracy to which the solution is obtained. Although the HHL algorithm originally exhibited quadratic dependence on $\kappa_A$, its poly-logarithmic dependence on the dimension of the problem opened the doors for a potential exponential quantum speedup for solving linear systems of equations.

Many linear systems problems (for example, solving the Newton system within an interior point method algorithm for linear or semidefinite optimization [7]) require high-precision solutions, which can be challenging to compute on imperfect hardware. Iterative Refinement (IR) is a classical technique originally developed in response to this challenge on low-precision classical computers. IR can use an inexact or low-precision solution as a starting point and systematically improve its accuracy [3, 9]. This idea has since been extended to compute highly accurate (and at times, *exact*) solutions to linear [1, 2] and semidefinite optimization problems [5], and recent works have investigated their adaptability to quantum algorithms [6, 8]. Given the speedups IR has facilitated for optimization, it is worthwhile to investigate how IR can be leveraged in combination with QLSAs. In this work, we show that both in theory and in practice, by using a NISQ device to solve for the residual error term at each step, we can achieve high-precision final solutions while requiring significantly fewer resources from the quantum hardware.

The iterative refinement method for solving linear system of equations [3] is described in Algorithm 1, and we summarize the core idea as follows. The algorithm commences from an initial solution $x^{(0)} \in \mathbb{R}^d$, and subsequently computes a refined solution $x^{(k+1)} \leftarrow x^{(k)} + u^{(k)}$ in iterations $k = 0, 1, 2, \ldots$, where $u^{(k)}$ acts as a correction of the error $r^{(k)} = z - Mx^{(k)}$ and is the solution to the *refining system* $Mu^{(k)} = r^{(k)}$. These operations can all be carried out using the same level of accuracy or *fixed precision*.

### 1.1   Contributions

Our primary contribution is a benchmarking suite, available at `https://github.com/QCOL-LU/QLSAs`, for running the HHL algorithm with iterative refinement on quantum computer simulators/emulators as well as on real hardware. The suite is integrated with the Quantinuum qnexus library for seamless execution on supported backends, including Quantinuum's own H-series hardware and

high-performance emulators. Our implementation is generalized (not hard-coded to specific instances or sizes), unlike the majority of available implementations found online.

We also demonstrate that classical IR post-processing significantly lowers the quantum resource requirements for HHL. By iteratively refining a low-precision quantum solution, we reduce the need for high-precision quantum phase estimation (QPE). This directly translates to shallower circuits with fewer gates, fewer qubits required for the QPE register, and viable execution on current NISQ hardware, effectively enabling a fault-tolerant-era algorithm to provide meaningful results today. Iterative refinement also reduces statistical noise in the estimate of the solution vector during tomography. This reduces the number of shots required to estimate a quantum state to a given level of precision.

Lastly, we benchmark results for systems up to 4 variables on Quantinuum H2-2, and 8 variables on various emulators/simulators, quantifying the resource reductions and demonstrating exponential improvements in solution accuracy even on real noisy hardware.

## 2 Algorithm Implementation

Here, we discuss the details of our HHL-IR implementation.

### 2.1 Iterative Refinement

Our implementation follows the fixed-precision IR scheme detailed in and summarized in Algorithm 1, which alternates between a quantum solver and classical post-processing. This approach uses the quantum computer only for the most challenging step (solving the linear system), while leveraging classical resources for all scaling and update operations.

---

**Algorithm 1** Iterative Refinement for the Linear Systems Problem

---

**Input**: Coefficient matrix $A \in \mathbb{R}^{d \times d}$ with $\left\|A^{-1}\right\| \leq \kappa_A$, right-hand side vector $b \in \mathbb{R}^d$, error tolerances $0 < \zeta \ll \xi < 1$.

**Output**: A classical solution vector $x \in \mathbb{R}^d$ satisfying $\left\|Ax - \frac{b}{\|b\|}\right\| \leq \zeta$.

Let $\tilde{\alpha} = \max\{\|A\|_F, \|b\|\}$. Then, normalize the system $\tilde{b} \leftarrow \frac{b}{\tilde{\alpha}}$ and $\tilde{A} \leftarrow \frac{A}{\tilde{\alpha}}$.
Choose starting point $(x^{(0)}, r^{(0)}, \eta^{(0)}) \leftarrow (0, \tilde{b}, 1)$, $k \leftarrow 0$.
**while** $\|r^{(k)}\| > \frac{\zeta}{\tilde{\alpha}}$

1. $\tilde{u}^{(k)} \leftarrow \textbf{solve}\ (A, \eta^{(k)} r^{(k)})$ using $O_{LS}(\xi)$
2. $x^{(k+1)} \leftarrow x^{(k)} + \frac{1}{\eta^{(k)}} \tilde{u}^{(k)}$
3. $r^{(k+1)} \leftarrow \tilde{b} - Ax^{(k+1)}$
4. $\eta^{(k+1)} \leftarrow \frac{1}{\|r^{(k+1)}\|}$
5. $k \leftarrow k + 1$

**end**

---

## 2.2   HHL Circuit and Execution

The core quantum component is the HHL algorithm, which we implement in Qiskit. The circuit consists of three main stages: Quantum Phase Estimation (QPE) to find the eigenvalues of the matrix A, a controlled rotation based on the inverted eigenvalues, and an inverse QPE to uncompute the register. For hardware execution on Quantinuum systems, circuits are transpiled using pytket and submitted via the qnexus library, which handles the full compilation and job management workflow. While most existing HHL implementations currently available are hardcoded to a given linear system, this is not sufficient for our IR scheme because the right hand side of the system changes at each iteration. Thus, our implementation flexibly accepts input matrices and vectors just as classical linear solvers do.

## 3   Experimental Results

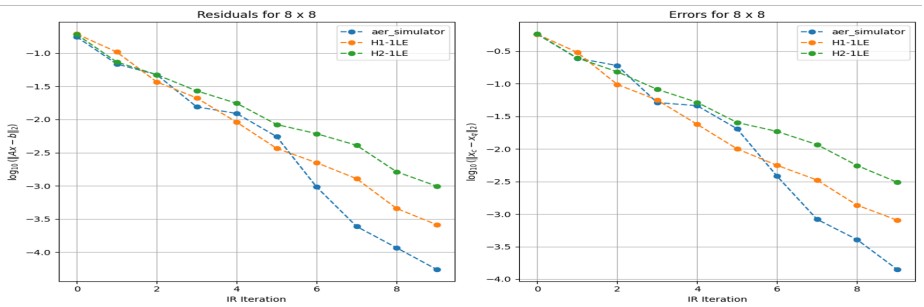

**Fig. 1.** Residuals and errors from an 8 variable linear system as a function of iterative refinement iteration on Quantinuum emulators and an IBM Qiskit simulator.

We run HHL-IR on various backends, including both emulators/simulators and quantum hardware. The results demonstrate that iterative refinement consistently and significantly improves the quality of the solutions obtained from the HHL algorithm.

Figure 1 shows the log-scale residuals and errors for an 8-variable linear system solved on Quantinuum emulators and an IBM Qiskit simulator. As the number of IR iterations increases, both the error and residual decrease exponentially, confirming the effectiveness of the refinement process in noiseless simulatation.

This trend is successfully reproduced on NISQ hardware. Figure 2 plots the results for a 4-variable system executed on the Quantinuum H2-2 quantum computer. Despite the presence of hardware noise, IR still exponentially reduces the solution error and residual, improving the accuracy of the final result with each iteration.

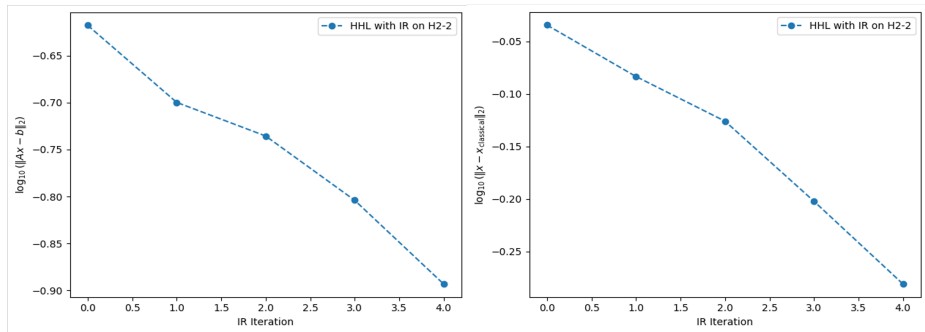

**Fig. 2.** Residuals and errors from a 4 variable linear system as a function of iterative refinement iteration on Quantinuum H2-2.

A detailed summary of our benchmarking is presented in the table in Table 1. This data quantifies the performance of HHL with and without IR across various problem sizes and backends. For every case, the application of IR yields a dramatic reduction in both the final solution error $||x_c - x_q||$ and the residual $||Ax - b||$. For example, for an 8-variable problem on the H1-1LE emulator, IR reduced the solution error by over three orders of magnitude, from approximately 0.8488 to 0.0008. Note that we set a maximum of 10 iterations in these experiments, but in some cases IR converges in less. These results empirically validate that IR is a crucial tool for obtaining high-precision solutions from quantum linear solvers on current and near-term hardware.

**Table 1.** Example statistics gathered in our QLSA benchmarking suite on a variety of problem sizes and backends.

| Backend | Problem Size | Condition Number | Sparsity | Number of Qubits | Circuit Depth | Total Gates | $\|\mathbf{x}_c - \mathbf{x}_q\|$ without IR | $\|\mathbf{x}_c - \mathbf{x}_q\|$ with IR | $\|A\mathbf{x} - \mathbf{b}\|$ without IR | $\|A\mathbf{x} - \mathbf{b}\|$ with IR | Total Iterations of IR |
|---|---|---|---|---|---|---|---|---|---|---|---|
| aer_simulator | 2 x 2 | 1.872340 | 0.500000 | 4 | 21 | 25 | 0.379778 | 0.000000 | 0.327757 | 0.000000 | 5 |
| H1-1LE | 2 x 2 | 1.872340 | 0.500000 | 4 | 12 | 22 | 0.255828 | 0.000050 | 0.121943 | 0.000042 | 10 |
| H2-1LE | 2 x 2 | 1.872340 | 0.500000 | 4 | 12 | 22 | 0.281076 | 0.000092 | 0.132106 | 0.000051 | 10 |
| aer_simulator | 4 x 4 | 2.571542 | 0.625000 | 7 | 107 | 132 | 0.547150 | 0.000030 | 0.141385 | 0.000009 | 9 |
| H1-1LE | 4 x 4 | 2.571542 | 0.625000 | 7 | 323 | 449 | 0.711828 | 0.096980 | 0.189305 | 0.024518 | 10 |
| H2-1LE | 4 x 4 | 2.571542 | 0.625000 | 7 | 323 | 449 | 0.765401 | 0.058367 | 0.204540 | 0.014552 | 10 |
| aer_simulator | 8 x 8 | 2.232571 | 0.750000 | 12 | 1007 | 1093 | 0.771855 | 0.000142 | 0.258965 | 0.000055 | 10 |
| H1-1LE | 8 x 8 | 2.232571 | 0.750000 | 12 | 2997 | 4310 | 0.848844 | 0.000807 | 0.305588 | 0.000260 | 10 |
| H2-1LE | 8 x 8 | 2.232571 | 0.750000 | 12 | 2997 | 4310 | 0.857951 | 0.003080 | 0.309835 | 0.000992 | 10 |

## 4 Conclusions

Quantum linear systems algorithms such as HHL can provide speedups over their classical counterparts. However, many instances of linear systems problems require solutions calculated to high precision. Although the requirements

for achieving these levels of precision directly with QLSAs like the HHL algorithm are out of reach on today's hardware, integrating QLSAs with iterative refinement can exponentially improve the quality of the solution with linear cost in time and no cost in space. Here, we close the gap between theory and practice, providing an open-source implementation with state of the art results in solution quality on current NISQ hardware.

Future work will focus on expanding this benchmarking suite to further enhance its practicality and scope. We plan to incorporate and evaluate other promising quantum linear systems algorithms, such as variational QLSAs, to provide a more comprehensive comparison of available methods. To continue reducing the resource overhead, we will also implement more efficient variations of the HHL algorithm for specific problem instances, like using the Hadamard test to avoid costly tomography procedures. Finally, we will extend backend support to a wider array of quantum platforms beyond Quantinuum and IBM, ensuring our findings are broadly applicable and robust across different hardware architectures.

**Acknowledgments.** This research used resources of the Oak Ridge Leadership Computing Facility, which is a DOE Office of Science User Facility supported under Contract DE- AC05-00OR22725.

**Disclosure of Interests.** The authors have no competing interests to declare that are relevant to the content of this article.

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

# Benchmarking Refined Quantum Linear Systems Algorithms

**Adrian Harkness[1], Kate Saltovets[1], Mohammadhossein Mohammadisiahroudi[1], Tamás Terlaky[1]**

[1] Lehigh University, PA, USA

## Quantum Linear System Algorithms

### Motivation

- Systems of linear equations are fundamental to science, engineering, and machine learning.
- Quantum Linear Systems Algorithms (QLSAs) like HHL offer a potential exponential speedup over classical methods.

### Challenges

- A significant gap exists between the theoretical promise of QLSAs and their practical implementation.
- Current Noisy Intermediate-Scale Quantum (NISQ) hardware has high error rates and limited resources.

### Our Contribution

- We present a benchmarking suite for the HHL algorithm enhanced with classical Iterative Refinement (IR).
- Our implementation is open-source and generalized (not hard-coded to specific problems).

### Applications

- Quantum Interior Point Methods
- Quantum Machine Learning

### Quantum Linear System Problem:

$$Ax = b$$

$$|b\rangle = \frac{\sum_{j=1}^{N} b_j |j\rangle}{\|b\|_2} \quad \boxed{\text{QLSA}} \quad |x\rangle = \frac{\sum_{j=1}^{N} x_j |j\rangle}{\left\|\sum_{j=1}^{N} x_j |j\rangle\right\|_2}$$

### Iterative Refinement:

**Algorithm**   Iterative Refinement for the Linear Systems Problem

**Input:** Coefficient matrix $A \in \mathbb{R}^{d\times d}$ with $\|A^{-1}\| \le \kappa_A$, right-hand side vector $b \in \mathbb{R}^d$, error tolerances $0 < \zeta \ll \xi < 1$.

**Output:** A classical solution vector $x \in \mathbb{R}^d$ satisfying $\left\|Ax - \frac{b}{\|b\|}\right\| \le \zeta$.

Let $\tilde{\alpha} = \max\{\|A\|_F, \|b\|\}$. Then, normalize the system $\tilde{b} \leftarrow \frac{b}{\tilde{\alpha}}$ and $\tilde{A} \leftarrow \frac{A}{\tilde{\alpha}}$.
Choose starting point $(x^{(0)}, r^{(0)}, \eta^{(0)}) \leftarrow (0, \tilde{b}, 1)$, $k \leftarrow 0$.
**while** $\|r^{(k)}\| > \frac{\zeta}{\tilde{\alpha}}$

  1   $\tilde{u}^{(k)} \leftarrow$ **solve** $(A, \eta^{(k)} r^{(k)})$ using

  2   $x^{(k+1)} \leftarrow x^{(k)} + \frac{1}{\eta^{(k)}} \tilde{u}^{(k)}$

  3   $r^{(k+1)} \leftarrow \tilde{b} - A x^{(k+1)}$

  4   $\eta^{(k+1)} \leftarrow \frac{1}{\|r^{(k+1)}\|}$

  5   $k \leftarrow k + 1$

**end**

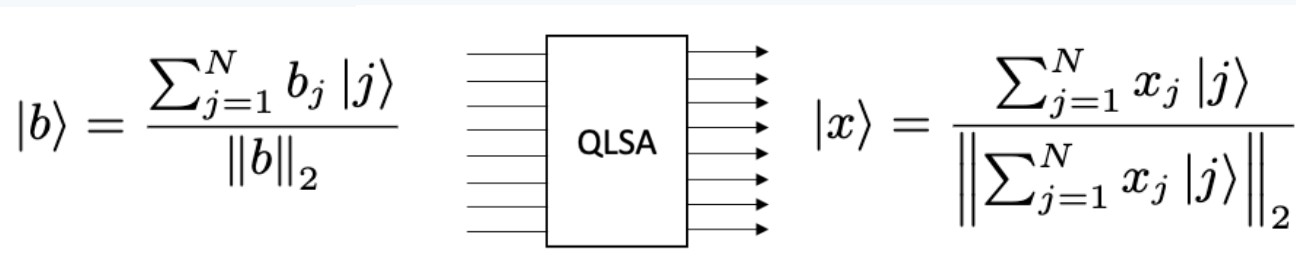

## Benchmarking HHL-IR

### Quantinuum H2-2:

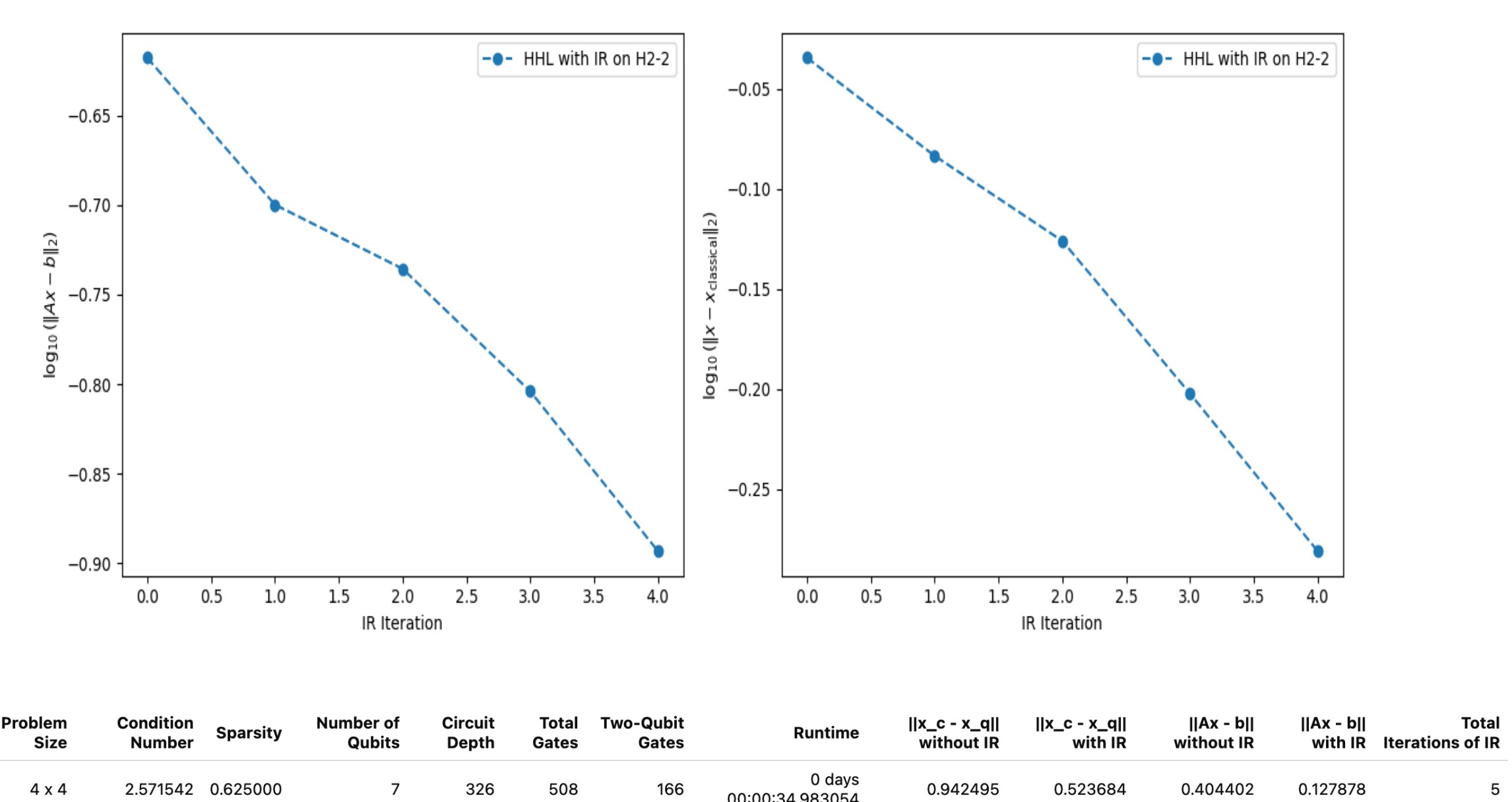

| Backend | Problem Size | Condition Number | Sparsity | Number of Qubits | Circuit Depth | Total Gates | Two-Qubit Gates | Runtime | \|\|x_c - x_q\|\| without IR | \|\|x_c - x_q\|\| with IR | \|\|Ax - b\|\| without IR | \|\|Ax - b\|\| with IR | Total Iterations of IR |
|---|---|---|---|---|---|---|---|---|---|---|---|---|---|
| H2-2 | 4 x 4 | 2.571542 | 0.625000 | 7 | 326 | 508 | 166 | 0 days 00:00:34.983054 | 0.942495 | 0.523684 | 0.404402 | 0.127878 | 5 |

### Projected performance on emulators:

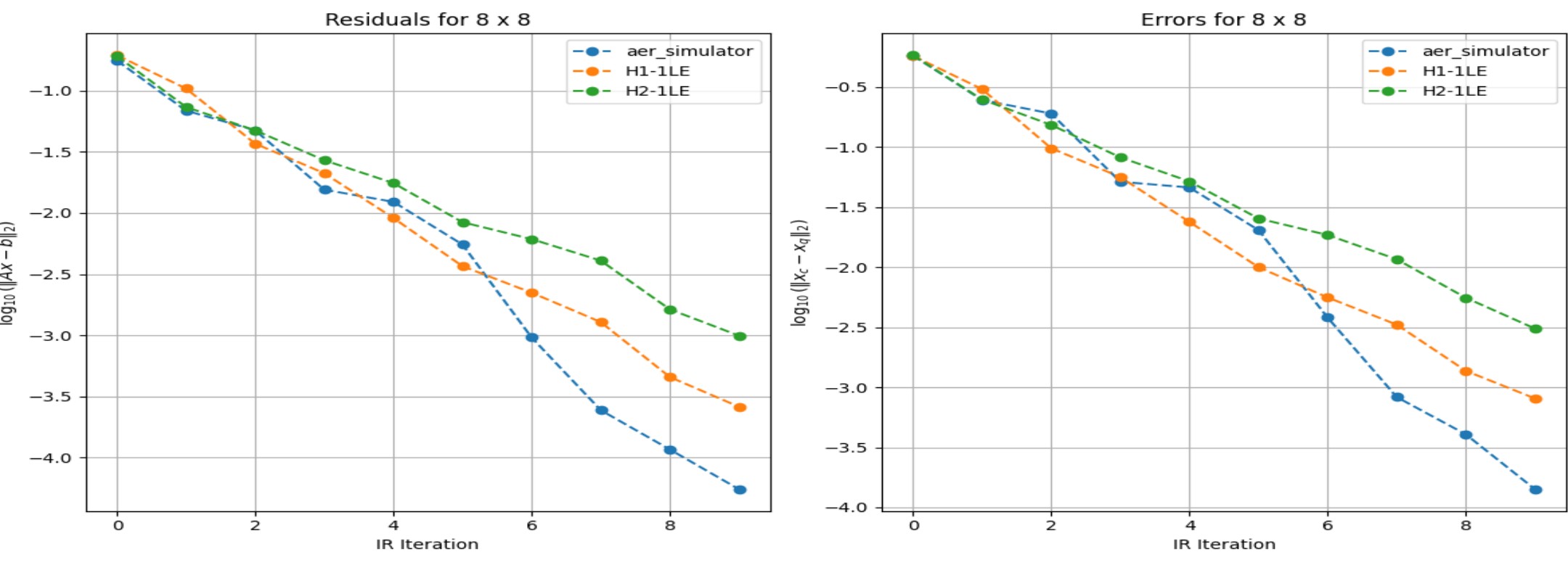

| Backend | Problem Size | Condition Number | Sparsity | Number of Qubits | Circuit Depth | Total Gates | \|\|x_c - x_q\|\| without IR | \|\|x_c - x_q\|\| with IR | \|\|Ax - b\|\| without IR | \|\|Ax - b\|\| with IR | Total Iterations of IR |
|---|---|---|---|---|---|---|---|---|---|---|---|
| aer_simulator | 2 x 2 | 1.872340 | 0.500000 | 4 | 21 | 25 | 0.379778 | 0.000000 | 0.327757 | 0.000000 | 5 |
| H1-1LE | 2 x 2 | 1.872340 | 0.500000 | 4 | 12 | 22 | 0.255828 | 0.000050 | 0.121943 | 0.000042 | 10 |
| H2-1LE | 2 x 2 | 1.872340 | 0.500000 | 4 | 12 | 22 | 0.281076 | 0.000092 | 0.132106 | 0.000051 | 10 |
| aer_simulator | 4 x 4 | 2.571542 | 0.625000 | 7 | 107 | 132 | 0.547150 | 0.000030 | 0.141385 | 0.000009 | 9 |
| H1-1LE | 4 x 4 | 2.571542 | 0.625000 | 7 | 323 | 449 | 0.711828 | 0.096980 | 0.189305 | 0.024518 | 10 |
| H2-1LE | 4 x 4 | 2.571542 | 0.625000 | 7 | 323 | 449 | 0.765401 | 0.058367 | 0.204540 | 0.014552 | 10 |
| aer_simulator | 8 x 8 | 2.232571 | 0.750000 | 12 | 1007 | 1093 | 0.771855 | 0.000142 | 0.258965 | 0.000055 | 10 |
| H1-1LE | 8 x 8 | 2.232571 | 0.750000 | 12 | 2997 | 4310 | 0.848844 | 0.000807 | 0.305588 | 0.000260 | 10 |
| H2-1LE | 8 x 8 | 2.232571 | 0.750000 | 12 | 2997 | 4310 | 0.857951 | 0.003080 | 0.309835 | 0.000992 | 10 |

### Conclusions

- Our hybrid quantum-classical method successfully produces high-precision solutions on current NISQ hardware, overcoming limitations from noise and resource constraints.
- Iterative Refinement exponentially improves solution accuracy. For the 8-variable problem on the H1-1LE emulator, IR reduced the solution error by over 1000x (from 0.8488 to 0.0008).
- This work helps close the gap between the theoretical promise of QLSAs and their practical application, making fault-tolerant era algorithms useful for meaningful results today.

### Future Work

- Expand Algorithm Portfolio: Incorporate and benchmark other promising quantum linear systems algorithms, such as Variational Quantum Linear Solvers (VQLS).
- Enhance Efficiency: Implement more resource-efficient variations of the HHL algorithm, for instance, by using the Hadamard test to avoid costly state tomography.
- Broaden Hardware Support: Extend the benchmarking suite to support a wider array of quantum hardware platforms beyond Quantinuum and IBM.

### Implementation

- ✓ Python Package benchmarking, resource estimation, and running QLSAs with IR on different backends: https://github.com/QCOL-LU/QLSAs

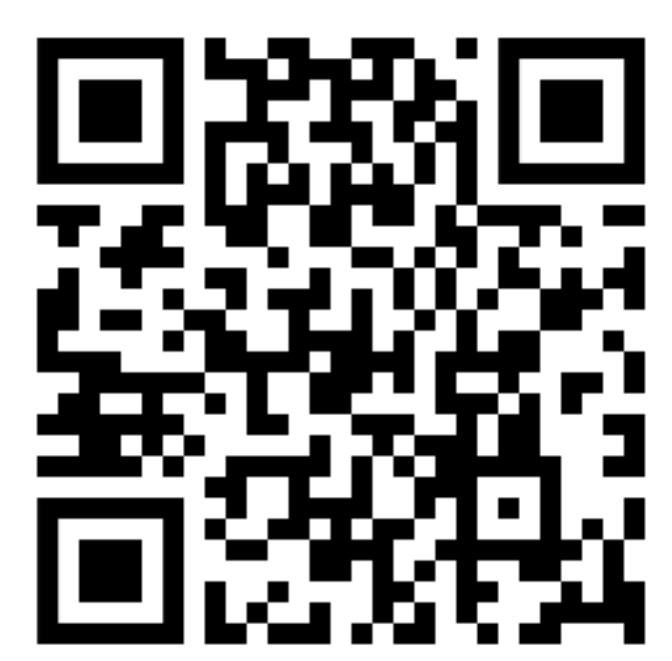

### Acknowledgements

- This research used resources of the Oak Ridge Leadership Computing Facility, which is a DOE Office of Science User Facility supported under Contract DE-AC05-00OR22725.

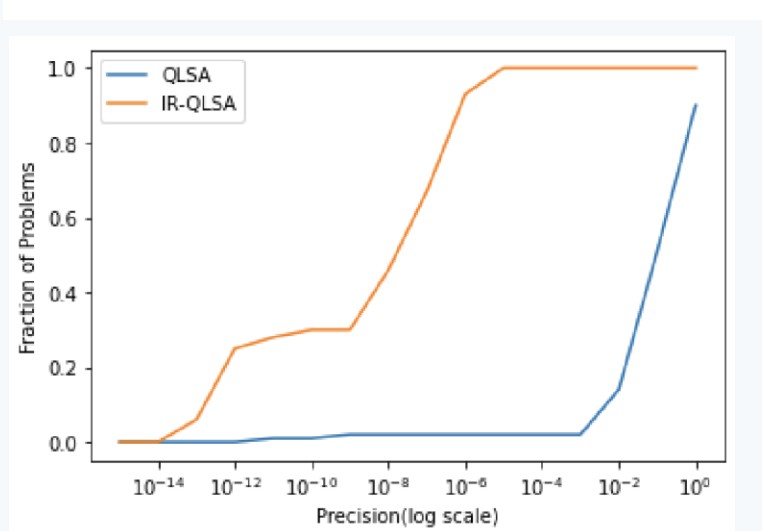
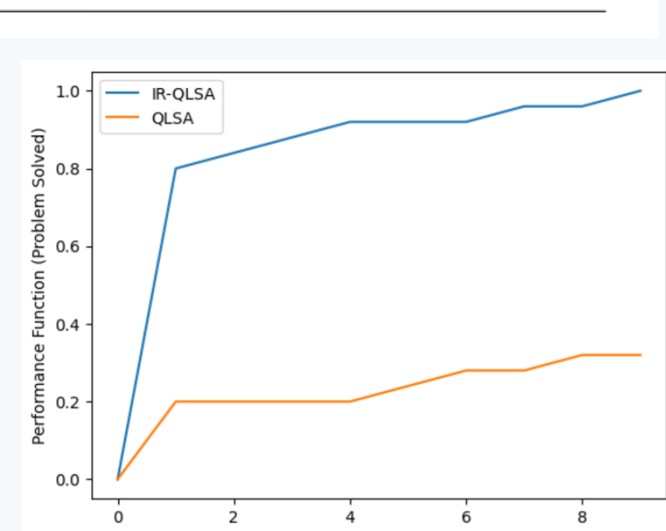