# OpenReview forum: "Benchmarking Refined Quantum Linear Systems Algorithms"
_purdue.edu/Purdue_University/PQAI/2025/Symposium — PQAI 2025 Oral_

### Official Review · Reviewer_5CHE · 2025-07-19

**Rating:** 6
**Confidence:** 4

**Review:**

A solid work with experiments on real quantum hardware, with source code and a detailed description of the experiments. I also like Table 1 very much.

In “In this work, we show that both in theory and in practice, by using a NISQ device to solve for the residual error term at each step, we can achieve high-precision final solutions while requiring significantly fewer resources from the quantum hardware” I don't like that I don't see the “in theory” aspect sufficiently fulfilled.

Further comments:

In my opinion, a new paragraph should start at “In this work, we show".

Figure 1 looks vertically compressed. That is not nice. The plots should be redrawn.

„in the table in Table 1“ -> „in Table 1“

„from approximately 0.8488 to 0.0008“ -> „from approximately 0.8 to 0.0008“ or „from approximately 0.85 to 0.0008“

Recommendation:

Accept (Poster)

---

### Official Review · Reviewer_YZiw · 2025-07-25

**Rating:** 7
**Confidence:** 4

**Review:**

This paper presents benchmark results for quantum linear system solvers utilizing Iterative Refinement on Quantinuum devices. The HHL algorithm is recognized for its potential exponential speedup in solving linear systems; however, this claim carries several caveats. For instance, the output is a quantum state rather than a classical vector, and applying tomography to the resulting quantum state yields only a low-accuracy solution. The central premise of the method investigated in this paper is the application of iterative refinement to exponentially reduce error. This paper's primary contribution is a software package implementing this method, along with experimental results demonstrating that this quantum-classical hybrid approach can effectively achieve high-accuracy solutions, even in the presence of quantum device errors. Nonetheless, certain implementation details are omitted from the paper. For example, a key limitation of the HHL algorithm, specifically how to implement and update block-encodings of the input matrix and how to prepare the vector state |b⟩, is not discussed. Despite these omissions, the experimental results are compelling.

I recommend its acceptance as a poster.

---

### Decision · Program_Chairs · 2025-07-29

Accept (Oral)